# In-situ measurements of the ice flow motion at Eqip Sermia Glacier using a remotely controlled UAV

Guillaume Jouvet<sup>1</sup>, Eef van Dongen<sup>1</sup>, Martin P. Lüthi<sup>2</sup>, and Andreas Vieli<sup>2</sup> <sup>1</sup>Laboratory of Hydraulics, Hydrology and Glaciology, ETH Zurich, Switzerland <sup>2</sup>Department of Geography, University of Zurich, Switzerland **Correspondence:** jouvet@vaw.baug.ethz.ch

**Abstract.** Measuring the ice flow motion accurately is essential to better understand the time evolution of glaciers and ice sheets, and therefore to better anticipate the future consequence of climate change in terms of sea-level rise. Although there exist a variety of remote sensing methods to fill this task, in-situ measurements are always needed for validation or to capture high temporal resolution movements. Yet glaciers are in general hostile environments where the installation of instruments

- might be tedious and risky when not impossible. Here we report the first-ever in-situ measurements of ice flow motion using a remotely controlled Unmanned Aerial Vehicle (UAV). We used a multicopter UAV to land on a highly crevassed area of Eqip Sermia Glacier, West Greenland, to measure the displacement of the glacial surface with the aid of an on-board differential GNSS receiver. Despite the unfortunate loss of the UAV, we measured approximately 70 cm of displacement over 4.36 hours without setting foot onto the glacier a result validated by applying UAV photogrammetry and template matching techniques.
- Our study demonstrates that UAVs are promising instruments for in-situ monitoring, and have a great potential for capturing short-term ice flow variations in inaccessible glaciers a task that remote sensing techniques can hardly achieve.

### 1 Introduction

- Glacial motion is a key process governing the advance or the retreat of glaciers. The accurate recording of ice flow is therefore crucial to calibrate models that can predict the future evolution of glaciers (e.g., Aschwanden et al., 2016). Among them, marine-terminating glaciers show usually fast ice flow (up to several meters per day) due to buoyant forces, highly crevassed texture in response to intense tension, and high temporal variability due to change in the subglacial hydrological system. To capture the dynamics of marine-terminating glaciers, it is therefore necessary to develop methods that can measure the ice flow of inaccessible glacial areas in high temporal resolution.
- Weekly, monthly or yearly average ice flow are commonly tracked by satellite images (Moon et al., 2012; Heid and Kääb, 2012), some data being directly available from general observational programs such as MEASURES (Joughin et al., 2018) or Greenland Ice sheet CCI (http://esa-icesheets-greenland-cci.org). Yet revisit times of observation satellites are usually too long

to capture variations at daily or subdaily resolution and in-situ validation data is mostly lacking. Alternatively, one can use laser scanning (Pętlicki and Kinnard, 2016), or interferometric radar (Riesen et al., 2011; Lüthi et al., 2016) to increase the temporal resolution of data, however, the spatial coverage of such ground-based instruments remains limited. In recent years, Unmanned Aerial Vehicle (UAV) photogrammetry by Structure-from-Motion Multi-View Stereo (SfM-MVS) has been used increasingly for the remote sensing of glacial motion at high spatial resolution (Immerzeel et al., 2014; Ryan et al., 2015; Jouvet et al., 2017;

Bhardwaj et al., 2016; Chudley et al., 2019).

While aforementioned remote sensing methods for ice flow monitoring has made significant improvements, in-situ measurements by the means of accurate GPS remain necessary for validation purposes or to capture short time scale ice movements. For instance, Sugiyama et al. (2015) captured the dynamical response of the calving front of Bowdoin Glacier to tide. Yet,

- the installation of GPS stations on ice presents several challenges. First, they are challenging to maintain due to the effects of possible snow falls or melt, and there exist a risk to slide and to fall into a crevasse after a certain lapse of time. For these reasons, ice flow records are mostly done in ablation areas to prevent against snow coverage, and GPS devices are installed on stakes positioned within the ice for sustainability during melt episodes. On the other hand, the most dynamical glacial areas (such as marine-terminating glaciers) are often the most crevassed zones making the installation of instruments dangerous,
- possibly costly (involving helicopter operations) when not impossible. To our knowledge, nobody has ever tried to automatize this task using UAVs.

In this paper we report the outcomes of the first-ever use of UAV for in-situ sensing of the glacial motion. In July 2018, we landed a quadcopter UAV on Eqip Sermia Glacier, West Greenland – a highly crevassed and fast-moving tidewater glacier, and measured the ice motion for a couple of hours thanks to a on-board differential GNSS receiver. In the meantime, we performed

- traditional UAV photogrammetry over the glacier, and processed the resulting ortho-images by template matching in order to cross-check the ice motion record with another technique. This paper is structured as follows: First, we shortly describe the study site (Section 2), and provide technical details about the two measurement methods we used: i) the traditional remote sensing method by UAV photogrammetry (Section 3) ii) the new in-situ method by UAV-carried GNSS receiver (Section 4). Then, we compare the results given by the two methods in Section 5, and make recommendations in Section 6 for improving
- the reliability of our approach in the future.

#### 2 Study site

Eqip Sermia Glacier (69°48'N, 50°13'W) is a marine-terminating glacier located in the west of the Greenland Ice Sheet (Lüthi et al., 2016), see Fig. 1a. The glacier discharges into the ocean through a 3-4 km wide calving front lying over shallow bedrock, where it features fast ice flow up to 14 m  $d^{-1}$ , and frequent calving activity. Due to its intense dynamics, Eqip Sermia

is extremely crevassed, and mostly inaccessible for in-situ sensing. In July 2018, field measurements were carried out for ten days to monitor remotely the ice dynamics and the calving activity of Eqip Sermia Glacier by terrestrial radar interferometry and UAV photogrammetry.

5

# 3 Remote sensing method

Before landing our multicopter UAV on the glacier for measuring in-situ the ice flow motion (Section 4), we performed photogrammetrical UAV flights to produce ortho-images and Digital Elevation Models (DEMs) of Eqip Sermia Glacier, identify an appropriate landing area, and make an initial estimate of the ice flow. For that purpose, we closely followed the approach described by Jouvet et al. (2017) and Chudley et al. (2019). We briefly describe the method in this section, and refer to the two aforementioned articles for more details.

#### 3.1 UAV equipment

As UAV for photogrammetrical flights we used a 2-meter-wide fixed-wing "Skywalker X8" equipped with Sony  $\alpha$ 6000 camera described in Jouvet et al. (2017). However, unlike Jouvet et al. (2017), we did not install any Ground Control Points (GCPs)

10 on the sides of the glacier. To ensure an accurate georeferencing of the photogrammetrical results, we used the direct method described in Chudley et al. (2019), i.e. the camera location of each picture was determined using an on-board differential carrier-phase GNSS receiver (the same as described in Section 4.2), which can deliver relative centimeter accuracy when combined with a second one (called base station), which is fixed on the ground.

#### 3.2 Surveying missions

In total, we performed three large-scale surveys of Eqip Glacier on July 6, 8 and 11. For each flight, the UAV was programmed to fly autonomously along parallel lines covering the terminus of the glacier and about 550 meters above the glacier. The UAV collected overlapping pictures with a ground sampling distance of 15 to 20 cm, an overlap of 95% in flight direction and 75% in cross flight direction. On July 11, we performed an additional low-altitude flight to refine the resolution by a factor of ~4 over a zone of interest, which encompassed the future landing area of the multicopter UAV (Section 4 and Figs. 1c, 1d, and 1e).

#### 3.3 SfM-MVS photogrammetry

The images collected during the surveying flights were processed by Structure-from-Motion Multi-View Stereo (SfM-MVS) using Agisoft PhotoScan software (http://www.agisoft.com/) to generate ortho-images and DEMs with a resolution of 25 cm and 50 cm, respectively (Fig. 1a). As Chudley et al. (2019), we used GNSS-supported aerial triangulation to georeference the

- 25 photogrammetrical products in a direct manner without any GCPs. For that purpose, we processed the logs of the carrier-phase GNSS receivers to deliver centimeter-accurate picture location relative to the base station (Section 4.2), whose the absolute location was measured accurately using a differential dual-frequency LEICA GPS receiver. To assess the georeferencing accuracy, we installed five GCPs next to the take-off and landing site, performed an additional survey of this area with exactly the camera and mission parameters we used for large-scale surveys, and compared the positions of the GCPs reconstructed by
- 30 SfM-MVS and measured by using the differential dual-frequency LEICA GPS receiver. As a result, we found an horizontal error between 23 and 45 cm with a standard mean deviation of 8 cm. This represents an absolute error of about 2 pixels, which

is slightly less accurate than the error estimate (about 1 pixel) reported by Chudley et al. (2019) with a similar UAV equipment but different surveying setting.

# 3.4 Template matching

Once the ortho-images and DEMs obtained, we used the Matlab toolbox ImGRAFT (http://imgraft.glaciology.net/) to derive 5 ice flow horizontal velocities by template matching (Messerli and Grinsted, 2015) from ortho-images of July 6, 8 and 11, see an example in Fig. 1b.

#### 4 In-situ method

#### 4.1 UAV equipment

UAV flights for in-situ measurements of the ice flow motion were conducted using a customized version of the "Enduro"
(www.droneshop.biz, Fig. 2), which is a quadcopter featuring low Kv motors, long propellers, and high battery capacity to maximize flight duration. Our UAV was equipped with the "Pixhawk 2" open-source autopilot (https://pixhawk.org/) running on "arducopter" firmware (http://ardupilot.org/ardupilot/). The latter allows several modes, from manual to fully autonomous flights that follow a pre-programmed mission script stored in the autopilot. For our application, we equipped our UAV with a LIDAR range finder to estimate the distance to ground accurately and allow for smooth autonomous landings, a First-Person

- View (FVP) with an on-board pointing-down camera to give the UAV operator a real-time view of the ground, and long-range telemetry and remote control receivers. The UAV was powered by two Lithium Polymer batteries (240 Wh in total). In this configuration, the power consumption of the UAV varies from 250 to 500 W when flying, allowing roughly 30 to 60 minutes of flight time according to the conditions met and the distance traveled. In our case, the amount of time necessary to fly from the UAV operator to the landing site and return represented less than 8 minutes. However, the batteries were also used to power
- the UAV instruments and an extra GNSS receiver (see Section 4.2) while recording the ice motion. To save energy during the measurements, a remote-controlled switch was installed to shut down unnecessary instruments such as the FPV and the LIDAR shortly after landing. In this saving mode, the UAV consumes  $\sim$ 5 W. Finally, spikes were installed under the four legs to prevent the platform from sliding over the ice (Figs. 1f and 2).

# 4.2 Differential GNSS receiver

The antenna of a second GNSS receiver to measure the ice motion was installed on the UAV next to the antenna used for navigation (Fig. 2). Here we used the single-frequency Emlid Reach receiver (https://emlid.com/reach/), which logs carrier phase data in order to facilitate high positioning accuracy. Key advantages of the Emlid Reach receiver are the low cost (about 300 USD), the lightweight (20 g) and the low power consumption (~1.2 W).

For data processing in differential mode this receiver ("rover") works in combination with a second one ("base"), which is 30 fixed on the ground. Differential carrier-phase positioning yield to centimeter accuracy (relative to the base station) as long as

5

the distance between the two (base and rover) remain under 10 km, the differential ionospheric delay being negligible for such a small distance (Chudley et al., 2019). Although the Emlid receiver can be used for Real-Time Kinematic (RTK) (i.e. providing the cm-accurate position of the UAV in real time), we used it only in Post Processed Kinematic (PPK) mode for simplicity, i.e. we downloaded the log files of the rover and the base station once the measurements were completed and processed them afterward via the open-source software RTKLIB (http://www.rtklib.com/rtklib.htm).

To assess the positioning accuracy, we performed a static test by leaving the UAV immobile on the ground for approximatively 6.5 hours on a stable off-glacier area, and monitored the variability of its position over time (Fig. 3a). We found that 95% of the recorded positions (after differential processing) were less than 1.1 cm horizontally and 1.6 cm vertically from the mean values. In what follows, we interpret these numbers as the positioning accuracy of our measurement instrument.

## 10 4.3 Identification of the landing spot

A suitable site to land the multicopter UAV and to measure the ice motion must fulfill the following requirements: i) be sufficiently flat to prevent the UAV from turning over, and ii) be in the line of sight of the operator and less than 2 km away to ensure a reliable connection with the remote controller, telemetry and FPV (Fig. 4). Due to the fragmented topography of Eqip Sermia Glacier (Figs. 1c and 1d), few sites met these criteria. We identified our landing site on the middle flowline of

15 Eqip Sermia Glacier and at 1.5 km of the closest glacier margin from the detailed DEM inferred from the July 11 surveying flight (Fig. 1e). As the last photogrammetrical flight and the one in-situ measuring the ice flow motion (Section 4.4) could not be carried out consecutively, we had to correct the position to account for the ice flow motion. For that purpose, we estimated the ice motion of the selected landing spot by applying template matching (Section 3.3) to the large-scale ortho-images from July 8 and 11.

# 20 4.4 Landing on Eqip Sermia Glacier

On July 12, we operated the multicopter UAV in autonomous mode to land at the location selected in Section 4.3. The UAV took off at 21:59:30 (local time) in no-wind and good weather conditions, travelled at a horizontal speed of  $\sim 10 \text{ m s}^{-1}$  over a 1.5 km distance and at a 100 m altitude difference from the operator to the landing site (Fig. 4). As the UAV was not capable of landing with high accuracy (the GPS used for navigation was not differential), our strategy was to adjust the trajectory of

- the UAV manually during the landing stage via the remote controller to fine-tune the touching point in line with the images provided by the FPV (Fig. 1f). As a result, the UAV landed 3:36 minutes after take-off approximately 3 meters from the targeted landing spot, but over a slope of  $\sim 25\%$  (Fig. 1e), with the result that the UAV tilted over onto two of its propellers. The UAV was left in this inclined position and the battery voltage was monitored via telemetry to determine the time at which the UAV should return before the battery capacity would no longer be sufficient for the flight back, see Appendix A. After 4.36 hours a
- 30 take-off was attempted. Unfortunately its tilted position caused the UAV to flip over, and become impossible to salvage. Shortly before this happened, the log files of the extra GNSS receiver were downloaded over Wi-Fi so that the measurements of the displacement of the UAV for the period between landing and mishap could be retrieved.

# 5 Results

The data of the UAV-carried GNSS receiver (once processed with the base station) indicates an horizontal displacement towards the south-west of approximately 70 cm in 4.36 hours, i.e.  $3.7 \pm 0.06$  m d<sup>-1</sup> approximatively 240° w.r.t. the north direction, clockwise (Fig. 3b). On the other hand, the remote sensing method (UAV photogrammetry and template matching) shows that the ice here moved by  $3.4 \pm 0.1$  m d<sup>-1</sup> in a south-west direction (~239° w.r.t. the north direction, clockwise) on average 5 between July 6 and 11 (Fig. 3b). Our error estimates are based on georeferencing errors of photogrammetrical products and the variability found with the two other displacement fields from July 6-8 and July 8-11. As a consequence, the two methods agree well in terms of magnitude (less than 5% of difference) and directions of the ice flow (Fig. 3b). The remaining discrepancy is most likely due to differences in the record periods of each method (diurnal variability).

- The key advantage of in-situ GNSS receivers is that they can determine the ice flow motion in much higher temporal 10 resolution and with greater accuracy than any remote sensing method. To compute the horizontal ice flow velocity, we first averaged the time series of the positioning data to get rid of the noise induced by measure uncertainties. Here we used a mean of a 1-hour time window, which corresponds to  $\sim 16$  cm of displacement, i.e., 15 times the uncertainty of the differential carrierphase GNSS-based positioning. The results show that the ice velocity varied from 3.5 to 3.8 m d<sup>-1</sup> during the measurement
- period (Fig. 3c). Furthermore, we found a slight vertical motion of  $\sim$ 4 cm (not shown) during the same time period, which is 15 close to the estimated error.

#### Recommendations 6

While the record of ice motion was successful, our strategy to retrieve the UAV safely was not. To understand the causes and make recommendations to improve our method, we analyzed in detail the log files of the UAV autopilot (those recorded by the telemetry), and the positions of the extra GNSS receiver. We found two potential causes of the loss of the UAV: i) manual 20 inputs during the landing stage via the remote controller were found to be more sensitive than expected; this diminished the accuracy of the pilot to adjust the touching point from FPV; ii) although the actual landing site was relatively close to the target (less than 3 m, Fig. 1e), this terrain was steeper than the targeted one and our UAV was not designed to land on such a slope. Therefore it is crucial in the future to improve the landing accuracy. This calls for some improvements of both the method and the platform.

25

Methodwise, preliminary photogrammetrical flights provided crucial information to identify a suitable landing spot, and we recommend that such flights continue to be performed before attempting any further landing mission. However, we advise against using FPV for manual adjustment of the touching point as it is subject to piloting inaccuracies. Instead, the FPV could be used to verify the position of the UAV relative to topographical features (e.g. crevasses, Fig. 1f), using an ortho-image obtained

from a preliminary photogrammetrical flight as a reference. It must be stressed that the FPV alone without any reference – 30 even with controlled axis of the camera – could not be used to identify the landing site as the images provided by the on-board camera do not reflect sufficiently well the local topography of the glacier's uneven and steep slope surface. As an alternative to FPV for accurate landing, we recommend instead to use a RTK-equipped UAV, which uses the second GNSS receiver as we

did, but obtains the correction from the base station in real-time (By contrast, here we used it for post-processing), resulting in highly accurate positioning capability. Using the same base station and GNSS receivers for both the reconnaissance and the landing flight would be another improvement as the method would not require any absolute reference point, the measurement of its location introducing some additional error. Furthermore, the two flights (reconnaissance and landing) are better operated consecutively with little delay so as to avoid additional positioning errors when updating the landing location with respect to

the ice motion.

Platformwise, we recommend using a flatter design than the one used in the present study (Fig. 2) with a low center of gravity so that the UAV is much less unlikely to turn over. Most importantly, we recommend using guards under each propeller, as these could have saved the UAV from the delicate position experienced here. This first experiment was performed conservatively in

- terms of energy usage, the high power capacity of our UAV having not been fully exploited (the actual flying time was less than 10% of its capacity). In fact, we could have let the UAV measure the ice motion for more than 10 hours before recalling the UAV, as the capacity at the end of this time would still suffice to power the return flight, see Appendix A. Fully shutting-down the UAV with the sole energy use by the GNSS receiver would have reduced the power consumption by 4 and would have increased the recording time by the same factors. Finally, doubling the battery capacity would be possible but this would result
- in highly increasing the energy consumption during the flying time period. Distance between launch and measurement sites is a critical parameter for guiding the choice of platform for such an application. If the flight time is rather short (if the distance and the elevation difference between the recording site and the operator are small), the UAV would be better optimized for payload capacity (i.e., with high Kv motors, short propellers) by carrying additional batteries and staying longer on the ice. Conversely, a UAV optimized for flight time (as the one used in this study) is better used for long distances and large differences
- in elevation. A platform similar to the one used here, but optimized as recommended, would be able to measure the ice motion for more than 48 hours, while being operated up to 5 km away in calm and 0°C conditions, such as those found at Eqip Sermia Glacier.

This initial attempt to measure the ice flow motion in-situ using a UAV should be seen as a first step toward a further automatized workflow that aims to increase the number of sampling points. In this perspective, UAVs could be used as a sole mean of transportation and deployment of GNSS receivers. Indeed, a single UAV can be used to deploy multiple GNSS receivers on the ice, considering that such a station is about ten times cheaper than a UAV and can transmit the data to the operator remotely. McGill et al. (2011) used a similar approach to track the drift of icebergs. Ideally, the dropping procedure will be combined with a method to recover GNSS receivers. However, automatization of the retrieval of objects by UAV remains a delicate task, and an active domain of research in robotics (e.g. Suarez et al., 2017).

#### 30 7 Conclusions and perspectives

We measured the ice flow motion of Eqip Sermia Glacier using two different methods: i) an established remote sensing method based on UAV surveying, SfM-MVS photogrammetry and template matching ; and ii) a new in-situ sensing method based on a remotely controlled UAV and a differential carrier-phase GNSS receiver. The measurement location is so heavily crevassed

that it is inaccessible, even by helicopter. The two methods agreed well in terms of magnitude (less than 5% of difference) and directions of the ice flow (Fig. 3b).

The approach presented in this study has great potential to measure the short-term variability in ice motion point-wise, especially on inaccessible terrain. Therefore, it could be used to investigate stick-slip events (Lipovsky and Dunham, 2016), the tidal signal of the ice flow at ocean-terminating glaciers (Sugiyama et al., 2015), or short-lived speed-up events caused by an sharp change in the subglacial hydrological system, as for instance due to the outburst of a supraglacial lake (Kjeldsen et al., 2014). Beyond this specific application, the technique developed may be used to deploy other sensors in-situ – such as weather or seismic stations (Podolskiy et al., 2016) – over sectors of the glacier that are not accessible. This will be of particular interest once the technique is automatized since it will allow us to increase significantly the number of sampling points while reducing

costs and human risk when compared to current in-situ manned methods.

#### Appendix A: Battery's state of charge

Our UAV and the on-board GNSS receiver were powered by two 6S Lithium Polymer batteries (22.2 V, 10 Ah in total), which were fully charged at take-off. In this configuration the voltage was maximal ( $\sim$ 25V, Fig. A1). The UAV battery monitoring system indicated that the 3:36 min-long flight consumed approximately 6% of the battery capacity (i.e., 13.6 Wh of the

20 222 Wh). As a result, the voltage dropped to 24.5 V after the landing on Eqip Sermia Glacier. During the recording time of the ice flow, the battery voltage was used as an indicator of the state of discharge (Fig. A1). In the case of 6S Lithium Polymer batteries, 21 V was used as a threshold value to indicate imminent full discharge. If the time evolution of the voltage during the recording period is extrapolated linearly (Fig. A1), the UAV could remain for about 10 hours on the ice before reaching 21.5 V, which is enough capacity for the return flight (expected to last ~3.5 minutes, the same as the first flight). Under these

circumstances, the total consumption would have been roughly 100 Wh (30 Wh flying and 70 Wh non-flying), which is about half the capacity of our two batteries. This below-average performance can be explained by the low temperatures, which must have impacted the battery capacities. Despite uninterrupted sunshine conditions, the UAV remained in the shade after landing, and stayed at a location where the temperature was close to  $0^{\circ}$ C. For this first experiment, we used a conservative voltage threshold value (23.25 V instead of 21.5 V). As a result, we triggered the take-off from Eqip Sermia Glacier 4.36 hours after

the landing (Fig. A1).

- Acknowledgements. The authors wish to acknowledge Martin Funk, Thomas Stastny and Andreas Wieser for their helpful comments on the manuscript, Andreas Bauder and Shin Sugiyama for helpful discussions on the differential GPS and the derivation of ice velocity data, and Yvo Weidmann for introducing the Emlid Reach GNSS receiver to us. We thank Andreas Bauder for processing the positions measured by the dual-frequency LEICA GPS receiver. We thank Peter King for constructing and customizing our "Enduro" UAV, the team of ETHZ's Autonomous Systems Laboratory for their advices, and Daniel Gubser for technical support, as well as Susan Braun-Clarke for editing the
- English. This research was supported by the Dr. Alfred and Flora Spätli Fund and the ETH Foundation "Grant ETH-12 16-2" (Sun2Ice Project) and the Swiss Polar Institute (2018 Polar Access Fund of Eef van Dongen).