# Peer review of "In-situ measurements of the ice flow motion at Eqip Sermia Glacier using a remotely controlled UAV"

_Geoscientific Instrumentation, Methods and Data Systems, 2019_

## Short Comment (SC1) · 18 May 2019

Hello, my name is Rama, I'm from Airlangga University Indonesia. Airlangga University has also published a journal about United States UAVs. UAV itself even can do more than manned aircraft did, such as reducing the risk of pilot being killed and its maneuver and endurance are not bounded by man's inability.

Please also note the supplement to this comment:
https://www.geosci-instrum-method-data-syst-discuss.net/gi-2019-6/gi-2019-6-SC1-supplement.pdf

**Supplement:**

[supplement omitted: unrelated document]

---

## Referee Comment (RC1) · Poul Christoffersen (Referee) · 22 Jun 2019

The manuscript by Jouvet et al. presents in situ UAV derived observations of ice flow using a remotely operated quadcopter while also using an autonomous fixed-wing UAV to map the glacier surface in advance. The authors describe the techniques of both methods before presenting their key findings, i.e. the in situ measurement of displacement with a GNSS receiver mounted on the quadcopter. The authors discuss the potential of using this method to make direct measurements.

The main finding is a proof of concept: that remotely operated UAVs can be used not only as means of aerial surveying, but as a strategy to deploy GNSS instruments on

the surface of inaccessible terrain such as the highly crevassed glacier in this case. The work is clearly described and the figures support the text well. I enjoyed reading this manuscript, and have mostly minor comments:

Page 1, abstract, I would recommend not mentioning the loss of a UAV in the abstract.

P1, 11, short-term is relative and an imprecise descriptor in this context. I think the potential of the in-situ method is to make 'continuous' measurements over a certain period, in this case 10 hours.

P1, 16, some glaciers flow as fast as 20 m per day. Maybe add "or even faster".

P2, 7-16, paragraph needs supporting references.

P3, section 3.1, a few more details would be useful here.

P3, 15, 'large' in large-scale survey is relative and imprecise. To most, the scale here would appear small. Delete 'large'.

P3, section 3.2, a few more details would be useful here as well, e.g. total length of survey lines, area covered, camera parameters.

P3, section 3.3, it worth describing how this was accomplished in the field. Some details about what's needed to do photogrammetry on the spot, i.e. in the field would be useful (e.g. computer requirements, work flow)

P4, 1-2, .... add: "and differences in the method of calculation".

P4, section 3.4, I would label this section 'feature tracking' or 'glacier velocity derivation'. It needs to be expanded a bit. When exactly was the surveys performed? How did you deal with shading? Error?

P4, 14, LiDAR. Really! What type? The value of this seems understated. Expand.

P5, 6, how far from the base station was this test?

P5 section 4.3, this section is to some degree a part of the results. You could move it

to section 5 and label it: 5.1. Identification of . . ...

P5 section 4.4, this section is to some degree a part of the results. You could move it to section 5 and label it: 5.2. First attempt to land on . . ...

P6 section 5, if above change is made, this would become: 5.3. Successful landing on. . ... the section should start by describing how the UAV was successfully landed.

P6, 13, is 1 hour needed? If shorter would be OK, maybe mention that.

P6, section 6, here you describe the photogrammetry as preliminary. What makes it 'preliminary'? Would errors be smaller and accuracy higher, if the photogrammetry had been fully executed in the field? Could the mishap have been avoided if the photogrammetry had been done differently? Some additional information is needed to fully convey the recommendations.

P7, 5, . . . with little delay. . .. Sure, but accurate photogrammetry takes a while. Here there is room for a discussion about the pros and cons, i.e. what level of accuracy is needed? And how is that achieved as fast as possible?

P7, 11, . . .. 10 hours. . .. This is a nice proof of concept. There is room to speculate how this can be taken forward.

---

## Referee Comment (RC2) · Anonymous Referee #2 · 26 Jun 2019

The manuscript is written in acceptable English but some minor edits are needed which I am sure the authors can figure out after careful rereads. I have several questions and suggestions:

1. "We used a multicopter UAV...": I see that it is a quadcopter. Use "quadcopter" in place of "multicopter".

2. "Despite the unfortunate loss of the UAV": Not needed.

3. The literature survey given in the introduction is inadequate and must include recently published literature (e.g., https://www.earth-syst-sci-data.net/11/579/2019/, https://www.the-cryosphere.net/9/1/2015/tc-9-1-2015.pdf,

[Figure]

**GID**

4. I would suggest to include description of UAV and flight planning parameters in tabulated form.

5. "...resolution by a factor of ∼4": Give the exact GSD.

6. "...we found an horizontal error between 23 and 45 cm...": 23-45 cm of error while comparing with 70 cm of in situ displacement can be a huge uncertainty!

7. The text "The UAV was left in this inclined position and the battery voltage ... become impossible to salvage. Shortly before this happened" is unnecessary and should be removed. The first paragraph of "Recommendations" section related with this detailing should also be omitted. This is a case-specific issue and as such, does not arise due to methodological failures.

8. I have several issues with the entire methodology, the purpose of this research, and the results. First, what was the actual error in your in situ measurement? Second, why is this study needed? Making a point measurement on glacier surface for 4 hours or less is not going to give any relevant information. Either deploy 10 UAVs simultaneously (which of course is logistically and economically not possible) to get area-wide measurements or simply use stake-based classic approaches if we really want in situ measurements. The whole purpose of UAVs is to use them as an aerial remote sensing platform which can bridge the gap between spaceborne and in situ measurements. Third, I am sure that you must be knowing how different glacier velocities can be in different hours of a day. This variation can further enhance across seasons. And on yearly scales, we can observe even more variability. It will really be a blunder to extrapolate 70 cm in 4.36 hours to 3.7 ±0.06 m d−1 and compare it with daily photogrammetric measurements! Moreover, if you report that the two methods show only less than 5% of difference, then what is the need of the in situ measurement

when it in any case is not feasible for a monitoring at relevant spatiotemporal scale! However, in this 5% difference, you are not considering (adding/subtracting?) the 23-45 cm of error in photogrammetric method and the error in the in situ method. Next you write: "The key advantage of in-situ GNSS receivers is that they can determine the ice flow motion in much higher temporal resolution and with greater accuracy than any remote sensing method." Why? If the difference is within 5%, I would on any given day prefer a remote sensing measurement covering wide area in lesser time! What is the use of high temporal resolution measurements if they are only for 4 hours and just for a single point?! The different facies of glacier show widely varying flows even during a day and such an in situ measurement is not going to give us any relevant information unless we are investigating a known particular case.

Although I acknowledge the efforts of the authors, owing to the methodological and conceptual issues with this work which I mentioned above, I really find majority of the recommendations and conclusions repetitive/case-specific and lacking in novelty. In short, I do not see this work as a conceptual advancement in UAV-based glaciological research simply because the reasoning for justifying the need and relevance of this work is not strong enough in my view.

---

## Author Comment (AC1) · 21 Jul 2019

We would like to greatly thank you for your comments and suggestions, which will help to improve the manuscript. Among them, your comments 6, 7, and 8 call for some answers, which we give below.

[Figure]

6. "... we found an horizontal error between 23 and 45 cm... ": 23-45 cm of error while comparing with 70 cm of in situ displacement can be a huge uncertainty!

⇒ Absolutely, and this demonstrates why capturing short-term ice motion (i.e. average over ∼4 h) by remote sensing is inaccurate. By contrast, our in-situ GPS has an accuracy of ∼1 cm, which is ∼20 times better than the remote sensing method. **We propose to highlight this argument in the revised version.**

7. The text "The UAV was left in this inclined position and the battery voltage ... become impossible to salvage. Shortly before this happened" is unnecessary and should be removed. The first paragraph of "Recommendations" section related with this detailing should also be omitted. This is a case-specific issue and as such, does not arise due to methodological failures.

⇒ From our analysis, it is likely that the UAV could have been recovered if it landed closer to the targeted point (e.g. using RTK GPS) preventing it to flip over, or if we had installed guards under the propellers. Therefore, this directly concerns the methodology, and it seems important to us to fully describe the outcomes of our tests (including the failure to recover) in order to propose more robust solutions, as we do in 'Recommendations'.

8. I have several issues with the entire methodology, the purpose of this research, and the results. First, what was the actual error in your in situ measurement?

⇒ We have given estimates of error locations (1.1 cm horizontally and 1.6 cm vertically) in Section 4.3. **This will be clarified in the revised version.** This corresponds to the specifications of the Emlid Reach GNSS receiver (https://emlid.com/reach/), as well as error estimates given by other users (e.g. ()). This error translates into an error of $0.06$ m/d in ice velocity when scaled to the 4.36 hours of recording time period as reported in the 'Result' Section.

Second, why is this study needed? Making a point measurement on glacier surface for

4 hours or less is not going to give any relevant information.

⇒ We fully agree that one point measurement on the glacier surface for ∼4 hours is very little data. However, the goal of this paper is not to make a glaciological analysis of the flow field of Eqip Sermia glacier, but instead to present and discuss the potential of a new method, which might lead to a greater dataset in the future, from a preliminary experiment. Our study must be understood as a proof-of-concept, which proposes a methodology – with large room for improvements – to place a light-weight sensor (not necessarily a GPS station) in an inaccessible or dangerous terrain remotely without setting foot on this terrain. Our study can therefore be of interest for glaciologists who need to place sensors on ice for a relative cheap price when compared to helicopter operations, or more broadly to geo-scientists who work on other hostile terrains.

Either deploy 10 UAVs simultaneously (which of course is logistically and economically not possible) to get area-wide measurements or simply use stake-based classic approaches if we really want in situ measurements.

⇒ In our recommendations, we suggest that a drop and recovery procedure for light-weight GPS stations is more appropriate to deploy multiple stations by UAV instead of landing single UAVs. Stake-based classical approach is impossible at Eqip Sermia as the glacier is mostly inaccessible (too crevassed, see Fig. 1c) similar to many other terminus of fast flowing glaciers. This is a key motivation of our study to deploy GPS stations remotely by UAVs.

The whole purpose of UAVs is to use them as an aerial remote sensing platform which can bridge the gap between spaceborne and in situ measurements.

⇒ Here we have a different opinion. While remote sensing applications remain the main market share in the drone industry, the part of 'cargo' applications has a great potential for urgent, short-range, and light-weight delivery **We propose to report**

**this argument in the revised version giving some examples of pilot projects and applications.** The goal of our paper is to show that UAVs should not be seen only as aerial remote sensing platforms, but can also be considered for in-situ sensing. This is of special interest for placing sensors over rough and inaccessible terrains.

Third, I am sure that you must be knowing how different glacier velocities can be in different hours of a day. This variation can further enhance across seasons. And on yearly scales, we can observe even more variability.

$\Rightarrow$ Short-term variability of the ice flow of tidewater glaciers has been observed multiple times (e.g. ). Minute-scale velocity responses to large iceberg calving events have been reported by , observed the tide modulation of the ice flow of a tidewater glacier, and one-day-long speed-up events have been captured by and for giving a few examples. Some processes like calving event, tide, or supra-glacial lake outburst might cause short-term variations in the ice dynamics, which in turn might affect calving. Therefore, one needs methods that can reliably capture the ice flow variability also at short time scales as well. **For clarification, we propose to focus the application of our method to tidewater glaciers in the revised version.** We fully agree that the seasonal variability is even more important, however, it is not our intention to propose a method to capture this as our method is anyway not designed for long record periods, and remote sensing methods perform very well for that purpose.

It will really be a blunder to extrapolate 70 cm in 4.36 hours to $3.7 \pm 0.06$ m/d and compare it with daily photogrammetric measurements!

$\Rightarrow$ $3.7 \pm 0.06$ m/d is the 'instant' velocity averaged during the 4.36 hours of recording, and it is not our intention to extrapolate this velocity to a longer period. Method-wise, it is much easier to infer ice velocity averaged over long periods with classical remote sensing methods (UAV or satellite) than capturing short-term ice velocity, as this can

be done only by in-situ measurements (below a certain time period). Here, we tackle the challenge of measuring the ice flow in short time scales.

⇒ In our paper, we compared the 'instant' in-situ record with a 3-day long photogrammetric-based measurement for validation purpose only. Ideally, this should have been done comparing exactly the same time period (3 days), unfortunately, we were not able to leave our UAV on ice for 3 full days. Therefore, we can not exclude that natural ice flow variability explains part of the discrepancy observed between the two methods. Yet it must be stressed that the variability mostly affect the speed of ice and to a lower extent the flow direction. Therefore, the good match between the ice flow directions of the two methods (one degree of discrepancy) provides a reliable validation. **This argument will be added to the revised manuscript.**

Moreover, if you report that the two methods show only less than 5% of difference, then what is the need of the in-situ measurement when it in any case is not feasible for a monitoring at relevant spatio-temporal scale!

⇒ This small percentage difference shows that we can not evidence high temporal variability of the ice flow in the present reported case. However, we can not exclude that one or several short-lived accelerations occurred between July 8 and 11 without being captured (as being asynchronous with the in-situ measurement time period).

However, in this 5% difference, you are not considering (adding/subtracting?) the 23-45 cm of error in photogrammetric method and the error in the in situ method.

⇒ The 23-45 cm of error in photogrammetric method is taken into account. However, as the error in ice flow velocity linearly decreases with the time period (the longer the time duration, the larger the displacement and the smaller the relative errors), the errors in ice velocity gets very small: 23-45 cm over 3 days produces an error of $\sim 0.06$ m/d, which is small compared to absolute speed ($\sim 3.7$ m/d).

Next you write: "The key advantage of in-situ GNSS receivers is that they can determine the ice flow motion in much higher temporal resolution and with greater accuracy than any remote sensing method." Why?

⇒ GNSS receivers can log typically at 10 Hz. However, due to positioning errors, we can not directly derive velocities from positions, but from average-in-time positions to reduce the noise. Therefore, the true temporal resolution is reduced to approximatively the time needed to move by the positioning error length (here the ice moves by ∼ 1 cm during ∼4 min).

⇒ Relative positioning errors are ∼1 cm with in-situ GNSS receivers, and 23-45 cm with UAV photogrammetry (with Post Processed Kinematic as used here). The error in ice flow velocity is the location error divided by the time of the recording period. Therefore, for a long time period (e.g. 3 days), the two methods perform similarly well as the two errors are small compared to the absolute ice speed recorded. By contrast, the in-situ GNSS approach has a clear advantage for short time periods as the location errors (and then the ice flow error) are much smaller.

⇒ Another advantage of GNSS receivers is that they can record the vertical motion of ice as well (e.g. in response to tide). **We will add this information in the revised manuscript.**

If the difference is within 5%, I would on any given day prefer a remote sensing measurement covering wide area in lesser time! What is the use of high temporal resolution measurements if they are only for 4 hours and just for a single point?! The different facies of glacier show widely varying flows even during a day and such an in situ measurement is not going to give us any relevant information unless we are investigating a known particular case.

⇒ For the present case of Eqip Sermia, we agree that the gain of having in-situ GPS (against remote sensing method) is very limited as the results do not evidence high temporal variability. Our method aims to be applied to glaciers, which show a significant variability of the ice flow at short time scales as illustrated in aforementioned references.

Although I acknowledge the efforts of the authors, owing to the methodological and conceptual issues with this work which I mentioned above, I really find majority of the recommendations and conclusions repetitive/case-specific (⇒ see our answer to 7.) and lacking in novelty.

⇒ To our knowledge, no one before has ever deployed a sensor in-situ on an inaccessible glacial area in a remotely controlled way with a UAV. The key novelty of our paper is not the UAV photogrammetry (it is now commonly used for glacier surveying) nor the measurement technique it-self (GPS stations are commonly used to record ice dynamics), but the deployment technique by UAV, which does not require to set a foot on ice. We agree that the amount of data we collected for this study is modest, however, our paper does not aim to do an in-depth glaciological analysis of the ice dynamics of Eqip Sermia glacier, but instead reports on the potential of a new technique for an instrumentation journal. **We suggest to clarify in the beginning of the paper that this is a proof-of-concept study, which focuses on the remote and unmanned deployment of GPS sensors on ice using UAVs.**

In short, I do not see this work as a conceptual advancement in UAV-based glaciological research simply because the reasoning for justifying the need and relevance of this work is not strong enough in my view.

⇒ **We suggest to revise our manuscript to strengthen "the reasoning for justifying the need and relevance of this work" as follows:**

- **Better emphasize the need (high temporal resolution data, validation) to perform in-situ surveys of the ice motion by the means on GPS stations in complementarity to remote sensing methods, and illustrate the glaciological added-value of in-situ measurements (high temporal data, vertical motion captured, high gain in accuracy, weather independent method, ...) against remote sensing methods, especially in the case of tidewater glaciers.**

- **Better illustrate that manned in-situ deployment of sensors on ice can be dangerous, costly, and even impossible when the glacier is too crevassed.**

- **Better evidence that remote sensing methods can not capture high temporal resolution data, which are relevant to study some processes of tidewater glaciers.**

- **Clarify that we used Eqip Sermia glacier for test only to assess and validate the method.**

**References**

T. R. Chudley, P. Christoffersen, S. H. Doyle, A. Abellan, and N. Snooke. High-accuracy uav photogrammetry of ice sheet dynamics with no ground control. *The Cryosphere*, 13(3):955–968, 2019.

SH Doyle, B Hubbard, Poul Christoffersen, Tun Jan Young, C Hofstede, M Bougamont, JE Box, and A Hubbard. Physical conditions of fast glacier flow: 1. measurements from boreholes drilled to the bed of store glacier, west Greenland. *Journal of Geophysical Research: Earth Surface*, 2018.

G. Jouvet, Y. Weidmann, M. Kneib, M. Detert, J. Seguinot, D. Sakakibara, and S. Sugiyama. Short-lived ice speed-up and plume water flow captured by a vtol uav give insights into subglacial hydrological system of bowdoin glacier. *Remote sensing of environment*, 217:389–399, 2018.

T. Murray, M. Nettles, N. Selmes, L. M. Cathles, J. C. Burton, T. D. James, S. Edwards, I. Martin, T. O'Farrell, R. Aspey, I. Rutt, and T. Baugé. Reverse glacier motion during iceberg calving and the cause of glacial earthquakes. *Science*, 2015.

D. Podrasky, M. Truffer, M. Fahnestock, J. M. Amundson, R. Cassotto, and I. Joughin. Outlet glacier response to forcing over hourly to interannual timescales, jakobshavn isbræ, greenland. *Journal of Glaciology*, 58:1212–1226, 12 2012.

S. Sugiyama, D. Sakakibara, S. Tsutaki, M. Maruyama, and T. Sawagaki. Glacier dynamics near the calving front of Bowdoin Glacier, northwestern Greenland. *Journal of Glaciology*, 61:223–232, 2015.

---

## Referee Comment (RC3) · Anonymous Referee #3 · 22 Aug 2019

Review of Jouvet et al., Geosci. Instrum. Method. Data Syst., August 2019

I wrote/submitted this review before reading the two other reviews to avoid any cross-influence.

Summary

In their paper, Jouvet et al. present an innovative way to measure glacier movement by landing a UAV on a glacier and recording his position through time using a dGPS antenna. This technique is thus able to measure glacier displacements in regions that are inaccessible or too dangerous to human beings (highly crevassed regions of fast glaciers for example). The authors present the deployment, the data processing, the

results of 4 hours of data acquisition and the strength and limitations of this method.

Although the results are not breathtaking, I think this paper is worth publication. Publishing this experience will save a lot of time to colleagues who want to perform similar measurements in the future. It also points to an alternative methods (dropping GPS units using UAV) that may have a higher potential. The paper is also well-written.

General comments

My single general comment relates to the need to convince the readers that, despite some strong limitations, the method may still prove useful for the community. In the introduction (2.7 to 2.16), several limitations of existing methods to measure surface displacements are listed. I was expecting the novel method to solve these issues. Not really the case. In fact this part of the introduction and the corresponding discussion could focus on the actual/specific glaciological process(es?) that could be better understood using this technique. For example, does the community need continuous measurements of glacier movement at temporally high resolution just upstream/before very large calving event? If yes, why? Currently I feel like the "classical" UAV method, applying image correlation to repeat high resolution imagery has a stronger potential but maybe it cannot reach the temporal resolution of some of the processes of interest? If yes, to be explained. To summarize, can the authors tell why this method can make a real contribution? Right now with only a few hours of data and the constrain that the landing site is only 1-2 km away from the operator I feel that the limitations are very strong and may hamper acquisition of glaciologically useful measurements.

Specific comments.

3.12. "Centimer accuracy" seems optimistic given that horizontal errors of 23 to 45 cm (3.31) are obtained. Reconcile the two statements. "Decimeter accuracy"?

4.4. "once the images and DEMs are obtained"

4.26. Why choosing a single-frequency (less precise) unit? Lower cost?

5.7. What was the distance of the UAV to the base station for this test? Similar to the distance to landing site on glacier?

5.10. I think the landing spot should also be selected for his scientific interest. Maybe obvious but worth mentioning (see general comment also).

5.22. Why so late (22h)? To limit wind? It is cooler at night so not so good for the battery. Was there a scientific reason behing this choice of the time of the day?

6.12. "measured"

6.14. Was the direction also stable through time?

6.22. maybe "ability" rather than "accuracy"

7.19. Can the authors quantify a bit "long" and "large"?

8.6. The problem with a subglacial lake drainage is that (as far as I know) the event cannot be exactly forecasted. So how can the operator decide when he will land his UAV?

8.9. Can the authors clarify what sort of automatization they have in mind?

Figure 1. I wonder whether a shaded relief map or a slope map will not be better to illustrate the quest for the best landing site. Did the authors look at them?